# New Normal in ITCZ and Its Role in Altering Agroclimatic Suitability for Rice Production

**Somnath Jha [1], Mourani Sinha [2,*] and Anupam Kumar [3]**

[1] West Bengal Police Service, Home & Hill Affairs Department, Balurghat 733101, West Bengal, India; somnath.jha@gmail.com

[2] Department of Mathematics, Techno India University, Kolkata 700091, West Bengal, India

[3] Centre for Climate Research Singapore, Meteorological Services Singapore, National Environment Agency, Singapore 228231, Singapore; anupam_kumar@nea.gov.sg

[*] Correspondence: mou510@gmail.com; Tel.: +91-93-5090-4194

**Abstract:** Intertropical Convergence Zone (ITCZ) primarily governs the convective rainfall potential of the summer monsoon in Asia. In the present study, non-parametric trend test with outgoing longwave radiation (OLR) for the summer monsoon period for the last 42 years (1980–2021) have been analyzed for ITCZ zone, representative zones of Hadley circulation and Walker circulation for exploring trend of the deep convection activity. Besides, various climatic variables like temperature (maximum, minimum, mean), precipitation, and cloud cover dataset are used for exploring trend in major rice growing regions of the world. The results indicate that there is a significantly decreasing trend of OLR in ITCZ zone during summer monsoon season. Contrarily, major rice growing regions of the world have witnessed a significantly increasing trend for the temperature parameter among all the zones. Rainfall and cloud cover have shown a typical trend i.e., increasing rainfall but decreasing cloud cover in the Southeast Asian and Maritime Continent rice growing regions. In rice suitable climate assessment, it has been found that the Maritime Continent rice growing region, the Indo-Gangetic Plain and the Southeast Asian rice growing regions have witnessed better rice suitable climates than other rice growing regions during the last 42 years (1980–2021).

**Keywords:** ITCZ; walker cell; hadley cell; monsoon; rice; OLR; cloud cover

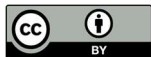

## 1. Introduction

Intertropical Convergence Zone (ITCZ) is a narrow band of deep convection playing an important role in the earth's climate. The outgoing longwave radiation (OLR) data can be used to represent the ITCZ [1]. The shift of the ITCZ impacts water availability in the tropical regions and thus affects climate change aspects [2]. The pole ward shift of the ITCZ increases the frequency of tropical cyclones [3] and quantum of Indian summer monsoon rainfall [4]. Ref. [5] discussed the behavior of regional ITCZs and regional monsoons and discussed in detail the annual variation of rainfall in the tropics and subtropics and the concept of global monsoon. Several studies have been done on the impact of climate change on rainfall pattern [6–8] but there are very limited studies on the impact of climate change on food availability [9–14] has studied the presence of strong regional circulation namely Hadley circulation zone over the Indian Ocean (HCIO) which has a deep impact on the dynamics of rainfall potential in summer monsoon. A teleconnection among the summer monsoon rainfall, ENSO (El Nino-Southern Oscillation) and IOD (Indian Ocean dipole) parameters and the rice production of certain regions has been studied recently [15]. Asia continent shares about 90% of the rice production across the globe [16,17] and 50% of the global population depends on rice as staple food [18]. There are various factors of crop production viz., crop seed variety or genetic characteristics of the crop, climate, terrain, soil properties, irrigation facility or rainfall sufficiency, nutrients

availability (fertilizer and manure management), insect-pest-pathogen management etc. [19]. Ref. [20] has studied in detail the yield gap analysis in rice crop in southwest China with multiple factors of production and reported that the rice yield gap has been found to be positively correlated with sunshine hour, phosphorus and potassium fertilizer and negatively correlated with soil available nitrogen content. Thus extensive studies and experiments have been conducted for assessing sensitivity of the rice crop to multiple factors including climatic and non-climatic factors. Ref. [21] has studied the impact of climatic factor in rice yield of Cambodia and it was reported that three climatic variables viz., maximum temperature, minimum temperature and rainfall explain approximately 63% and 56% of rice yield variability in wet and dry season, respectively and they further reported that the contributions of climatic factor and non-climatic factor in rice yield variability of Cambodia are 60% and 40%, respectively. Besides, the air temperature is the main factor among climatic factors impacting the growth of rice crop [22] because the source of air temperature is solar radiation which also controls the sunshine hour and nutrient mobility in soil. The impact of cloud cover on satellite images of rice producing areas of China has been discussed in the context to maximize its production [23]. In another study, rice yield variability of Asia due to climate change and ozone layer depletion has been studied [24]. Thus few studies have been conducted on the impact of climatic and non-climatic factors on rice yield on regional scale. In the context of climate change scenario, there are many standardized management practices to cope with other non-climate factors of production of rice and those standardized management practices are well within the ambit of research & development of modern agricultural sciences. Comparatively, climatic factors are exerting unprecedented challenges in climate change scenario where neo normal of the climatic variables has been challenging the long term mean or pattern of those variables. In reality, many rice growing regions are found with suitable or favourable (or non- suitable/unfavourable) climatic factors but with non-suitable or unfavourable (or suitable/favourable) non-climatic factors and vice versa. In practice, it is easy to ameliorate the unfavourable non-climatic factors vis-à-vis a favourable climatic factors applying various scientific and technical innovative approaches like fertilizer and manure management, irrigation management, integrated nutrient and pest management (INPM) techniques, biocontrol, precision farming, improved or genetically engineered planting materials or seed etc. But in case of unfavourable climatic factors it is not so easy to ameliorate the negative impact of climatic factors even in presence of favourable non-climatic factors. This inequality of impact of factors of production is also present in rice crop. Even though, various novel techniques for irrigation facility like drip irrigation, sprinkler irrigation, flood or splash irrigation techniques, zero tillage techniques etc may ameliorate the impact of less rainfall but mitigating the impact of air temperature is not convincing one in rice crop. The unfavourable climatic factors trigger a gradual change in genetic makeup of the rice crop as part of its climate adaptation tendency. This complexity of impact of unfavourable climate on rice production thus poses a serious concern for natural adaptation in rice crop biology in the near future. Thus the rice cultivars of various zones of the world under climate change scenario are being exposed to various degree of climatic adaptation. Therefore, there is a requirement of assessing the trend of climatic suitability only of rice crop production in major rice producing zones irrespective of its present rice crop production in a varied interplay of climatic and non-climatic factors. Thus the present study has been limited its scope with assessment of the role of climatic factors on rice and the agro-climatic suitability of rice has been assessed based on climatic parameters only. No study has been done yet on the major rice producing zones of Asia to investigate the pattern and trend of deep convective activity in ITCZ and its relation with major climatic variables in rice. The present study explores the relationship between the OLR parameter representing the ITCZ and the various climatic variables like temperatures (maximum, minimum, mean), precipitation, cloud cover for major rice growing regions. Broad physiological suitability criteria of agro-climate of rice growth has been assessed in this study.

## 2. Materials and Methods

In the present study, outgoing longwave radiation (OLR) data from NOAA Climate Data Record (CDR) having 2.5° × 2.5° spatial resolution (accessed from https://www.ncei.noaa.gov/products/climate-data-records/outgoing-longwave-radiation-monthly) (accessed on 24 December 2022) has been accessed and processed for the ITCZ spatial belts for summer monsoon months i.e., June, July, August and September (JJAS) as a season and individual monthly average for the period from 1980 to 2021.

Several reference studies have indicated a zone spanning from 10° N to 30° N latitude and from 20° W to 110° E longitude as the ITCZ spatial belt during the period of JJAS in which the ITCZ conventionally lies. Referral studies [25] indicated that the Walker circulation has typical four spatial loci on surface as zones which play important roles in the dynamics of Walker circulation. These zones are named as (i) Equatorial East African Region (EAR) (spanning from 3° S to 3° N latitude and from35° E to 45° E longitude), (ii) Congo Basin Region (CBR) (spanning from 3° S to 3° N latitude and from 10° E to 35° E longitude), (iii) Maritime Continent Region (MCR) (spanning from 15° S to 15° N latitude and from 90° E to 150° E longitude) and (iv) Equatorial Central Indian Ocean Region (CIOR) (spanning from 3° S to 3° N latitude and from 60° E to 90° E longitude). The above four zones of Walker Circulation have been selected for this present study along-with one Hadley circulation zone over the Indian Ocean (HCIO) (spanning from 25° S to 10° N latitude and from 60° E to 100° E longitude) for assessing trend of the deep convectional activity. JJAS mean OLR has been processed and analyzed for the spatial average over the above mentioned ITCZ zone, EAR, CBR, MCR, CIOR and HCIO for the last 42 years from 1980 to 2021 and the time series have been analyzed for determining the Mann Kendall trend analysis. Rice is grown in Asia primarily in China, India, Bangladesh, Myanmar, Indonesia and Pakistan. In general, this primary rice producing zone of Asia contributes more than 90% rice production of the world. The primary rice growing zones are broadly defined in the present study as (i) South East Asia Rice Growing Region (SEAR) (10° N–30° N latitude and 95° E–140° E longitude), (ii) North East Asia Rice Growing Region (NEAR) (30° N–54° N latitude and 120° E–144° E longitude), (iii) South_Asia_Rice Growing Region (SAR) (6° N–31° N latitude and 66 E°–95° E longitude), (iv) Maritime Continent Region (MCR) (6° N–9° N latitude and 95° E–140° E longitude). Besides these four zones, Indian subcontinent has been broadly divided into two zones namely (v) Indo Gangetic Plain Region (IGP) (21° N–28° N latitude and 75° E–87° E longitude) and (vi) Peninsular India Region (PI) (8° N–19° N latitude and 72° E–84° E longitude). Climatic variables like maximum temperature (Tmax), minimum temperature (Tmin), average temperature (Tmean), precipitation and cloud cover data were accessed from CRU TS 4.06 dataset [26] (accessed from https://catalogue.ceda.ac.uk/uuid/e0b4e1e56c1c4460b796073a31366980) (accessed on 24 December 2022). The gridded Climate Research Unit (CRU) monthly time series (TS) data are available on high resolution grid (0.5° × 0.5° grid) for the period from 1901 to 2021 for all the above climatic variables. The data were accessed and seasonal mean JJAS of all the climatic variables (except rainfall) were processed for the six rice growing zones (SEAR, NEAR, SAR, MCR, IGP and PI) computing spatial average of the grids within the zone for the four months (JJAS) season for the period of 42 years from 1980 to 2021It is noted here that JJAS sum values for precipitation are analyzed for the 42 years from 1980 to 2021 for those six rice growing zones. Mann Kendall Trend analysis has been done for all these climatic variables for the above six zones for analyzing the presence or absence of any significant monotonic trend of the climatic variables in those regions. Figure 1 depicts the various rice growing regions of world as considered in the present study.

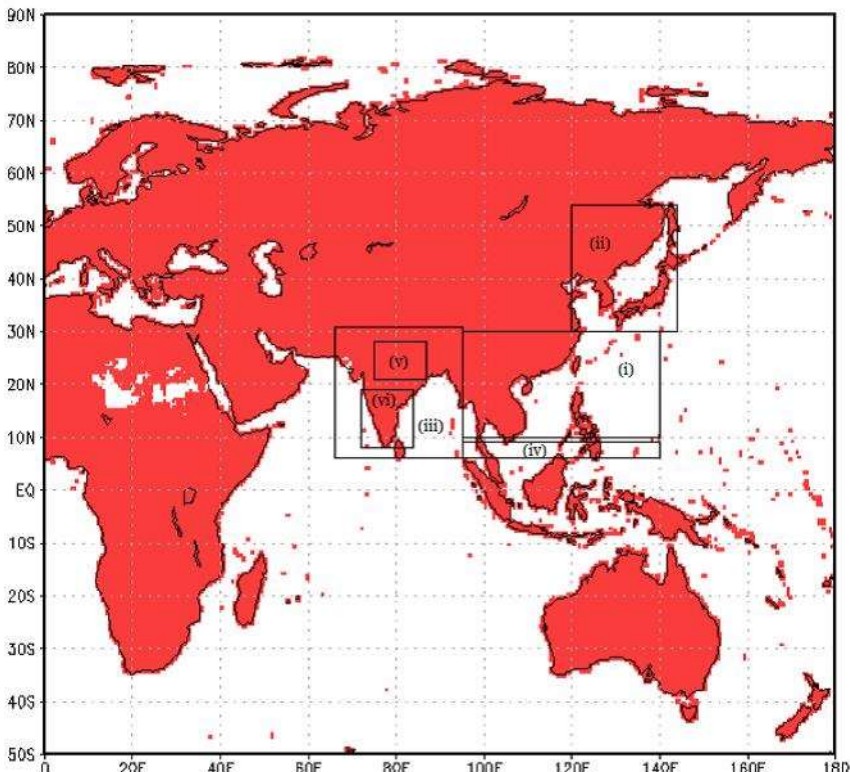

**Figure 1.** Major rice growing regions of world and its broad zones in boxes depicting (i)SE Asia Rice Growing Region (SEAR) (10° N to 30° N latitude and 95° E to 140° E longitude), (ii) NE Asia Rice Growing Region (NEAR) (30° N to 54° N latitude and 120° E to 144° E longitude), (iii) South Asia_Rice Growing Region (SAR) (6° N to 31° N latitude and 66° E to 95° E longitude), (iv) Maritime Continent Region (MCR) (6° N to 9° N latitude and 95° E to 140° E longitude). Indian subcontinent has been broadly divided into two zones namely (v) Indo Gangetic Plain Region (IGP) (21° N to 28° N latitude and 75° E to 87° E longitude) and (vi) Peninsular India Region (PI) (8° N to 19° N latitude and 72° E to 84° E longitude).

Nonparametric trend tests are extensively used for trend analysis for climatic variables and Mann Kendall test is such a nonparametric test for skewed-distributed atmospheric variables [27–32] has shown that power of nonparametric Mann Kendall test is higher than the power of parametric t-test. Well-tested DOS executable program for Mann Kendall Trend Test and Regional Kendall Trend Test have been taken from the United States Geological Survey website (accessed in http://pubs.usgs.gov/sir/2005/5275/downloads/) (accessed on 24 December 2022) for the present analysis. Finally, the significance test of output of the trend analysis was judged with two categories; highly significant with $p \le 0.01$ and significant with $0.01 < p \le 0.05$.

Agro-climatic suitability of rice is assessed purely based on temperature and rainfall physiological suitability criteria for this present study. The concept of cardinal temperature of rice has been used for the temperature suitability. Cardinal temperature is primarily the maximum, minimum and optimum temperature range within which the seed of particular species germinate and grow. Rice has such typical cardinal temperature as basal temperature (Tbase) 10 °C and Maximum Temperature (Tmax) 38 °C. Besides, 100 cm isohyet has been recognized since long as the constraint of rice in rainfed regions. 100 cm rainfall is at least required to grow rice under rainfed system and thereby a region with 100 cm isohyet can be easily regarded as climatically suitable for rice cultivation. Therefore, the following three broad conditions of climatic suitability have chosen to mark rice suitable agro-climate in the present study:

(1)   Rainfall Suitability: rainfall ≥ 1000 mm;

(2) Temperature Suitability: (a) Minimum Temperature ≥ 10 °C, and (b) Maximum Temperature ≤ 38 °C

The strike of JJAS or seasonal rainfall availability 1000 mm or above and air temperature within 10–38 °C are recorded for the agro-climatic suitability of rice growth.

In case of fulfillment of all above three conditions the JJAS season of the time series would be judged as rice suitable climate else non suitable condition. This logic has been developed to mark rice suitable climate or non-suitable climate for the 42 years (1980–2021) for all the six zones. In the present study, 0 and 1 logic has been used to mark for each year for rice non-suitable and suitable climate, respectively for calculating total score of suitability. Thus the rice suitable and non-suitable tests were conducted and a matrix has been prepared with the sum of such '1' i.e., rice suitable climates score for each of the six zones.

## 3. Results and Discussion

In the present study, non-parametric Mann Kendall Trend test have been performed for the JJAS average OLR activity for the ITCZ, HCIO (Hadley), EAR, CBR, MCR, CIOR (Walker) zones. Table 1 gives the statistics and the significance of the Mann Kendall Trend test performed on the OLR parameter for the above zones for summer monsoon season (JJAS) together. Table 2 gives similar analysis for the June, July, August, September months individually for ITCZ. Significant decreasing trend of OLR activity has been found in ITCZ and the decreasing trend is contributed to the significant decreasing OLR during the two months namely August and September. The comparison of rate of change in trend in Table 3 shows that the significant maximum rate of decreasing OLR activity in the ITCZ belt has been noted during August month. Table 4 accounts for the monthly analysis of the HCIO zone without any significance. Figure 2 depicts the significant decreasing trend of OLR for ITCZ.

**Table 1.** Mann Kendall Trend Test of OLR in W/m2 for JJAS.

| Zones | Tau Corr.Coeff | S | Z | *p* | Significance |
|---|---|---|---|---|---|
| ITCZ | −0.256 | −220 | −2.374 | 0.0176 | Decreasing |
| HCIO | 0.052 | 45 | 0.477 | 0.6335 | Non-Significant |
| EAR | −0.031 | −27 | −0.282 | 0.7781 | Non-Significant |
| CBR | −0.049 | −42 | −0.444 | 0.6568 | Non-Significant |
| CIOR | 0.052 | 45 | 0.477 | 0.6335 | Non-Significant |
| MCR | −0.049 | −42 | −0.444 | 0.6568 | Non-Significant |

**Table 2.** Mann Kendall Trend Test of OLR in W/m2 for ITCZ monthly.

| Month | Tau Corr.Coeff | S | Z | *p* | Significance |
|---|---|---|---|---|---|
| June | −0.020 | −17 | −0.173 | 0.8623 | Non-Significant |
| July | −0.105 | −90 | −0.965 | 0.3348 | Non-Significant |
| August | −0.238 | −205 | −2.211 | 0.0270 | Decreasing |
| September | −0.218 | −188 | −2.027 | 0.0427 | Decreasing |

**Table 3.** Trend equation for the significant trend of seasonal (JJAS) and monthly OLR activity in ITCZ belt.

| ITCZ Significantly Decreasing OLR | Tau Corr.Coeff | S | Z | *p* | Equation of Trend |
|---|---|---|---|---|---|
| JJAS Season | −0.256 | −220 | −2.374 | 0.0176 | $y = 384.26 + −0.06155\ x$ |
| August | −0.238 | −205 | −2.211 | 0.0270 | $y = 441.68 + −0.09269\ x$ |
| September | −0.218 | −188 | −2.027 | 0.0427 | $y = 413.57 + −0.07438\ x$ |

**Table 4.** Mann Kendall Trend Test of OLR in W/m2 for HCIO zone monthly.

| Month | Tau Corr.Ceff | S | Z | p | Significance |
|---|---|---|---|---|---|
| June | −0.022 | −19 | −0.195 | 0.8453 | Non-Significant |
| July | 0.039 | 34 | 0.358 | 0.7206 | Non-Significant |
| August | −0.041 | −35 | −0.368 | 0.7125 | Non-Significant |
| September | 0.073 | 63 | 0.672 | 0.5016 | Non-Significant |

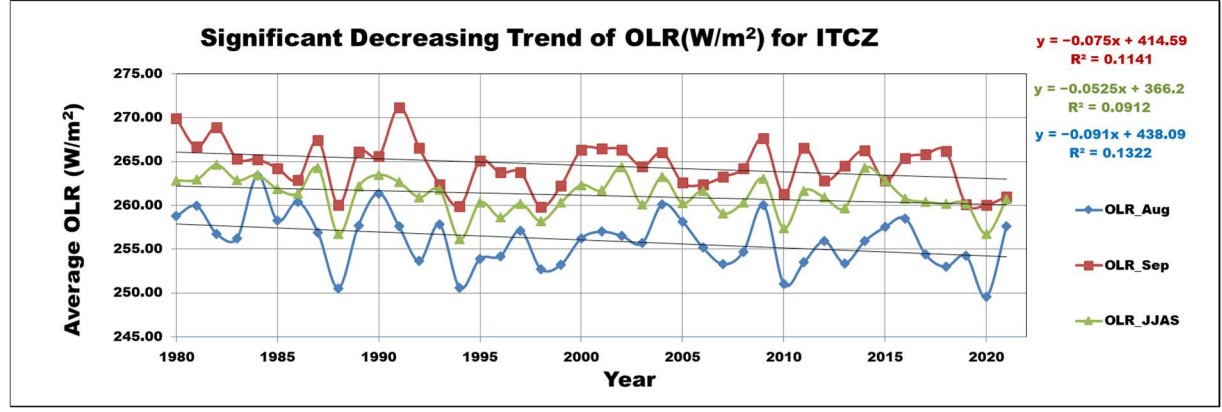

**Figure 2.** Significant decreasing trend of OLR for ITCZ.

Next Table 5 shows that no zone of Walker Cell region except CBR has any decreasing OLR activity.

**Table 5.** Mann Kendall Trend Test of OLR in W/m² for Walker zones monthly.

| Zones | Tau Corr.Coeff | S | Z | p | Significance |
|---|---|---|---|---|---|
| **Months** | | **EAR Zone** | | | |
| June | −0.071 | −61 | −0.650 | 0.5155 | Non-Significant |
| July | −0.049 | −42 | −0.444 | 0.6568 | Non-Significant |
| August | 0.120 | 103 | 1.105 | 0.2690 | Non-Significant |
| September | −0.116 | −100 | −1.073 | 0.2833 | Non-Significant |
| **Months** | | **CBR Zone** | | | |
| June | −0.215 | −185 | −1.994 | 0.0461 | Decreasing |
| July | 0.064 | 55 | 0.585 | 0.5584 | Non-Significant |
| August | 0.044 | 38 | 0.401 | 0.6884 | Non-Significant |
| September | 0.088 | 76 | 0.813 | 0.4163 | Non-Significant |
| **Months** | | **CIOR Zone** | | | |
| June | 0.043 | 37 | 0.390 | 0.6964 | Non-Significant |
| July | −0.036 | −31 | −31 | 0.7451 | Non-Significant |
| August | 0.008 | 7 | 0.065 | 0.9482 | Non-Significant |
| September | 0.048 | 41 | 0.433 | 0.6647 | Non-Significant |
| **Months** | | **MCR Zone** | | | |
| June | 0.006 | 5 | 0.043 | 0.9654 | Non-Significant |
| July | −0.094 | −81 | −0.867 | 0.3859 | Non-Significant |
| August | 0.091 | 78 | 0.835 | 0.4040 | Non-Significant |
| September | −0.041 | −35 | −0.368 | 0.7125 | Non-Significant |

Next Mann Kendall Trend test is performed on various climatic parameters of rice growing regions. The next six tables correspond to the Mann Kendall trend test for climatic variables viz., maximum temperature (Tmax), minimum temperature (Tmin), mean

temperature (Tmean), precipitation (Precip), cloud cover (Cloud_Cov) for rice production regions of the world as discussed previously. Table 6 corresponds to the SEAR zone for the parameters Tmax, Tmin, Tmean, Precip and Cloud_Cov. Tables 7–11 correspond to the NEAR, SAR, MCR, IGP and PI zones for the analysis of the given climatic variables.

**Table 6.** Mann Kendall Trend Test of JJAS averaged climatic variable for SEAR zone.

| SEAR Zone | Tau Corr.Coeff | S | Z | *p* | Significance |
|---|---|---|---|---|---|
| Tmax | 0.354 | 305 | 3.296 | 0.0010 | Increasing |
| Tmin | 0.515 | 443 | 4.793 | 0.00001 | Increasing |
| Tmean | 0.448 | 386 | 4.176 | 0.00001 | Increasing |
| Precip | 0.233 | 201 | 2.167 | 0.0302 | Increasing |
| Cloud_Cov | −0.482 | −415 | −4.488 | 0.00001 | Decreasing |

**Table 7.** Mann Kendall Trend Test of JJAS averaged climatic variable for NEAR zone.

| NEAR Zone | Tau Corr.Coeff | S | Z | *p* | Significance |
|---|---|---|---|---|---|
| Tmax | 0.401 | 345 | 3.728 | 0.0002 | Increasing |
| Tmin | 0.474 | 408 | 4.412 | 0.00001 | Increasing |
| Tmean | 0.453 | 390 | 4.217 | 0.00001 | Increasing |
| Precip | 0.029 | 25 | 0.260 | 0.7948 | Non-Significant |
| Cloud_Cov | −0.245 | −211 | −2.276 | 0.0228 | Decreasing |

**Table 8.** Mann Kendall Trend Test of JJAS averaged climatic variable for SAR zone.

| SAR Zone | Tau Corr.Coeff | S | Z | *p* | Significance |
|---|---|---|---|---|---|
| Tmax | 0.369 | 318 | 3.438 | 0.0006 | Increasing |
| Tmin | 0.420 | 362 | 3.913 | 0.0001 | Increasing |
| Tmean | 0.407 | 350 | 3.783 | 0.0002 | Increasing |
| Precip | 0.161 | 139 | 1.496 | 0.1348 | Non-Significant |
| Cloud_Cov | 0.261 | 225 | 2.428 | 0.0152 | Increasing |

**Table 9.** Mann Kendall Trend Test of JJAS averaged climatic variable for MCR zone.

| MCR Zone | Tau Corr.Coeff | S | Z | *p* | Significance |
|---|---|---|---|---|---|
| Tmax | 0.352 | 303 | 3.275 | 0.0011 | Increasing |
| Tmin | 0.527 | 454 | 4.912 | 0.00001 | Increasing |
| Tmean | 0.489 | 421 | 4.556 | 0.00001 | Increasing |
| Precip | 0.215 | 185 | 1.994 | 0.0461 | Increasing |
| Cloud_Cov | −0.352 | −303 | −3.273 | 0.0011 | Decreasing |

**Table 10.** Mann Kendall Trend Test of JJAS averaged climatic variable for IGP zone.

| IGP Zone | Tau Corr.Coeff | S | Z | *p* | Significance |
|---|---|---|---|---|---|
| Tmax | 0.251 | 216 | 2.331 | 0.0197 | Increasing |
| Tmin | 0.251 | 216 | 2.331 | 0.0198 | Increasing |
| Tmean | 0.262 | 226 | 2.440 | 0.0147 | Increasing |
| Precip | 0.099 | 85 | 0.910 | 0.3626 | Non-Significant |
| Cloud_Cov | 0.397 | 342 | 3.696 | 0.0002 | Increasing |

**Table 11.** Mann Kendall Trend Test of JJAS averaged climatic variable for PI zone.

| PI Zone | Tau Corr.Coeff | S | Z | *p* | Significance |
|---------|----------------|-----|--------|--------|------------------|
| Tmax | 0.288 | 248 | 2.678 | 0.0074 | Increasing |
| Tmin | 0.341 | 294 | 3.177 | 0.0015 | Increasing |
| Tmean | 0.315 | 271 | 2.928 | 0.0034 | Increasing |
| Precip | 0.178 | 153 | 1.647 | 0.0995 | Non-Significant |
| Cloud_Cov | −0.171 | −147 | −1.583 | 0.1135 | Non-Significant |

Mann Kendall Trend tests of various rice growing regions have shown a significantly increasing trend for the Tmax, Tmin, Tmean among all the zones. The rainfall and cloud cover parameters have shown a typical trend i.e., increasing rainfall but decreasing cloud cover in two zones i.e., SEAR and MCR zones where the rainfall is gradually increasing but the cloud cover is decreasing with time. The gap between rainfall and cloud cover is widening with time in SEAR and MCR regions (Figures 3 and 4). This matter requires to be looked into detail because the increasing trend of rainfall is significant but the gradient is gradual in compared to the steep decreasing gradient of cloud cover in these two regions. Both the regions like Southeast Asia and the Maritime Continental regions lie in tropics. Here the decreasing cloud cover but increasing rainfall indicates the every possibility of decreasing the formation of low cloud or raincloud during the summer monsoon season.

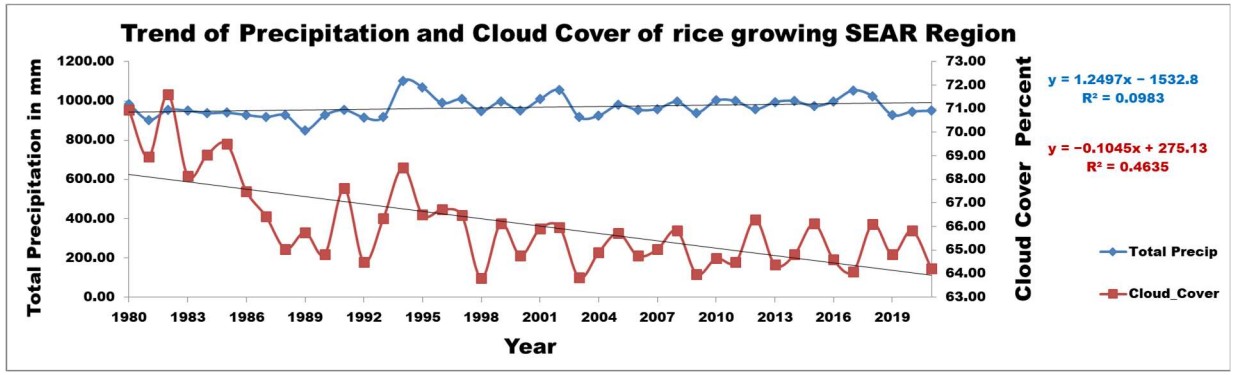

**Figure 3.** Trend of Precipitation and Cloud Cover for SEAR region.

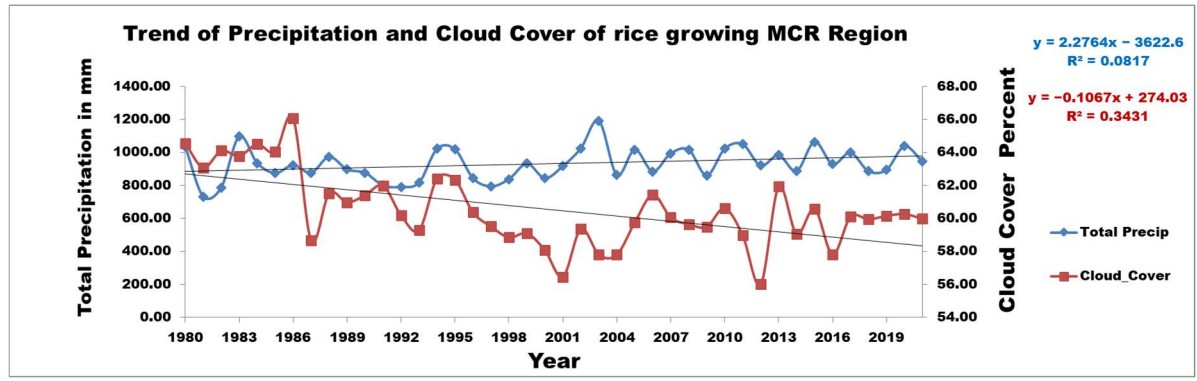

**Figure 4.** Trend of Precipitation and Cloud Cover for MCR region.

Rice suitable and non-suitable climate trends have been assessed based on simple rule of cardinal temperatures [22] lying within base temperature (Tbase) 10° C and maximum temperature (Tmax) 38 °C and precipitation 100 cm or more. The above rice climate suitable condition has been screened year-wise for 42 years and the score of the same has been prepared. It has been found that MCR and IGP zones followed by SEAR zone have

witnessed the most rice suitable climate with MCR and IGP zones facing the best score in rice climate suitability. (Table 12 and Figure 5)

**Table 12.** Rice Suitable Climate Year Score for 42 years (1980–2021) for various rice producing zones.

| Zones | SEAR | NEAR | SAR | MCR | IGP | PI |
|---|---|---|---|---|---|---|
| Rice Suitable Climate | 8 | 0 | 1 | 12 | 12 | 6 |
| Rice Non Suitable Climate | 34 | 42 | 41 | 30 | 30 | 36 |

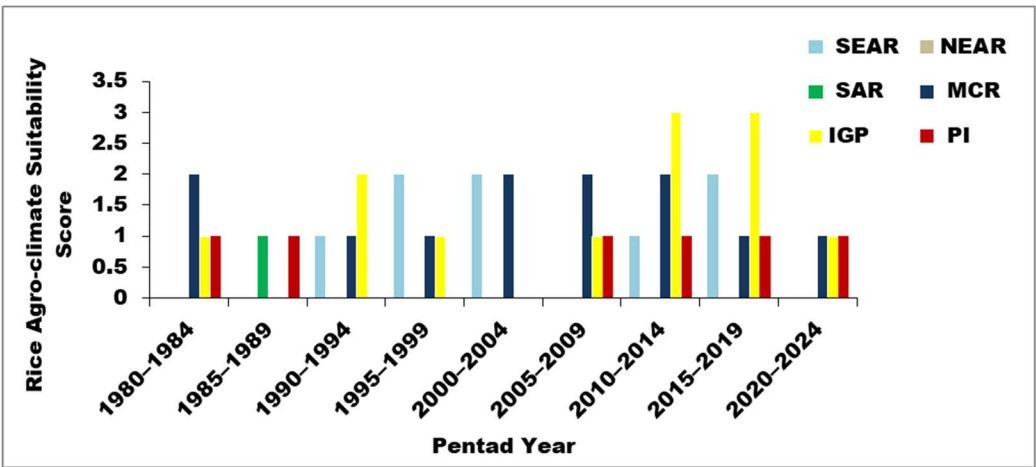

**Figure 5.** Rice suitability score in various pentad years from 1980 to 2021 for various rice growing regions of world.

Rice crop has adapted to various climatic zones with its changing genotypes. But in general rice has a biologically climatic suitability range of condition for growth. Rice has typical cardinal temperature for its viability of growth.

The Table 12 has shown that the MCR and the IGP have the highest such rice suitable climate score with 12 such years fulfilling all the conditions of rice growing out of total 42 years (1980–2021). Results showed that the next best rice suitable climate score has been taken by the SEAR with 8 and PI with 6 such years out of 42 years and the NEAR has seen the worst result with non- suitable condition. Northeast China falls within this NEAR zone of rice production and experiments on rice genotypes especially on tropical *japonica* subspecies has revealed to adapt a wide variability of cardinal temperatures and climatic condition and this relevant genetic pool is being used for breeding to evolve new cultivar for such climatic adaptability.

Figure 5 and Table 13 have shown that MCR and IGP regions have consistently distributed rice suitable climate score during the last few pentad years. Results have also indicated a typical trend in PI zones. SAR and NEAR zone have hardly found suitable agro-climate for rice by this pentad year wise strike score. This is indicating towards decreasing dependency on rainfed rice cultivation across the zone and increasing dependency on the irrigated rice. The typical peninsular topographic relief with broad oceans on both sides and the typical wide variability of *japonica* accessions may indicate towards this consistent suitable climate in regular interval of pentad. Table 14 also shows the trend of this suitable and non-suitable score across the four zones where the rate of change has been found to be high for IGP followed by SEAR.

**Table 13.** Rice suitable climate score for various rice growing regions of the world in pentad years from 1980 to 2021.

| Pentad Year | SEAR | NEAR | SAR | MCR | IGP | PI |
|---|---|---|---|---|---|---|
| 1980–1984 | 0 | 0 | 0 | 2 | 1 | 1 |
| 1985–1989 | 0 | 0 | 1 | 0 | 0 | 1 |
| 1990–1994 | 1 | 0 | 0 | 1 | 2 | 0 |
| 1995–1999 | 2 | 0 | 0 | 1 | 1 | 0 |
| 2000–2004 | 2 | 0 | 0 | 2 | 0 | 0 |
| 2005–2009 | 0 | 0 | 0 | 2 | 1 | 1 |
| 2010–2014 | 1 | 0 | 0 | 2 | 3 | 1 |
| 2015–2019 | 2 | 0 | 0 | 1 | 3 | 1 |
| 2020–2021 | 0 | 0 | 0 | 1 | 1 | 1 |

**Table 14.** Trend equation of rice agro-climatic suitability/non-suitability for SEAR, MCR, IGP and PI zones.

| Rice Growing Zones | Trend Equation | R² |
|---|---|---|
| SEAR | $y = 0.0667\,x + 0.5556$ | 0.0387 |
| MCR | $y = 0.0333\,x + 1.1667$ | 0.0167 |
| IGP | $y = 0.1833\,x + 0.4167$ | 0.2017 |
| PI | $y = 0.05\,x + 0.4167$ | 0.075 |

**4. Conclusions**

In summary, there is significantly decreasing trend of JJAS average OLR in ITCZ for JJAS as season and the later months of summer monsoon i.e., August & September contribute towards this decreasing trend. Therefore, August and September months of the summer monsoon season have witnessed deep convective activity in the ITCZ during the last 42 years (1980–2021). None of the Hadley Centre zone or Walker cell zones had any significant trend in OLR activity and thus it indicates that convectional activities along these spatial zones have not undergone significant change. Contrarily, major rice growing regions of the world have witnessed a significantly increasing trend for the Tmax, Tmin, Tmean among all the zones. The rainfall and cloud cover have shown a typical trend i.e., increasing rainfall but decreasing cloud cover in two zones i.e., SEAR and MCR zones where the rainfall is gradually increasing but the cloud cover is decreasing with time. In rice suitable climate assessment, it has been found that the MCR, IGP zones followed by SEAR zone have witnessed rice suitable climates during the last 42 years (1980–2021). These climatic suitable-prone zones of major rice growing area are thus exposed to less stress and indicate towards a sustainable rice producing belt for future whereas the rest of the zones, although are contributing towards production of rice, indicate non-sustainable rice producing zones with high degree of climatic adaptation in the near future.

**Author Contributions:** Conceptualization, S.J., M.S. and A.K.; methodology, S.J., M.S. and A.K.; software, S.J.; validation, S.J., M.S. and A.K.; formal analysis, S.J., M.S. and A.K.; investigation, S.J., M.S. and A.K.; resources, S.J., M.S. and A.K.; data curation, S.J.; writing—original draft preparation, S.J. and M.S.; writing—review and editing, S.J., M.S. and A.K.; visualization, S.J.; supervision, A.K.; project administration, S.J.; funding acquisition, S.J. and A.K. All authors have read and agreed to the published version of the manuscript.

**Funding:** This research received no external funding.

**Data Availability Statement:** Data used in this study are available from the corresponding author on reasonable requests.

**Acknowledgments:** The authors are thankful to NOAA (https://doi.org/10.7289/V5W37TKD) and CRU (https://crudata.uea.ac.uk/cru/data/hrg/cru_ts_4.06/) (accessed on 24 December 2022) for the datasets.

**Conflicts of Interest:** The authors declare no conflict of interest.

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
