# Peer review of "New Normal in ITCZ and Its Role in Altering Agroclimatic Suitability for Rice Production"

_climate, doi:10.3390/cli11030052_

Round 1

Reviewer 1 Report

This looks like an interesting analysis that would be of interest to the readers of this journal, however the manuscript is poorly constructed and difficult to understand. More details are needed in the methodology to understand the analysis process.  Tables full of insignificant stats are not helpful.  Figure 4 looks more suited to text statements.  On page 8 we read several times that rice has a minimum cardinal temperature of 100^C, I think a decimal point might be missing.  I would encourage the authors to spend more time carefully editing the manuscript for clarity and to remove errors before resubmitting.  As it stand the issues with presentation are too numerous to enable a fair judgement of the scientific content. 

Reviewer 2 Report

This manuscript examines how climatic changes alter the conditions of rice production. The manuscript is well structured, and the field is presented clearly, even if sometimes need some typographic improvements. The topic is suitable to be published in this journal, even if there are some minor revisions and one major revision:

Major revision:
The sentence on lines 220-229 needs some improvements. Is it possible to know if the sustainability will became better or deteriorate in the future? The authors calculate some trends with Mann Kendall test, but then decide to use an on/off summary (table 11), that erases any trend information. In my opinion it is very intriguing to know the rate of the changes and try to make some prediction for the future.

Minor revisions:
- line 29: playing a major role. Not clear, perhaps "important"?

- line 45: air temperature is a major factor. Perhaps "main"?

- line 57: JJAS. This acronym should be defined here.

- line 58: walker circulation: walker should be capitalized as in line 57.

- line 58: dy-nam-ics. In all the manuscript sometimes dashes interrups words in a wrong way. Please check the text.

- lines 220-233: Basal temperature 100 C and Maximum temperature 380 C. Probabily the authors intend 10°C and 38°C as reported in lines 213-214.

Round 2

Reviewer 1 Report

The paper has addressed the concerns raise in the first review, however it could still be improved with editing for grammar and clarity.

Reviewer 2 Report

This is the second version of the manuscript. Authors have satisfied the requests of my review. In particular, in the introduction they explained more clearly the topic and the problems that could affect the rice cultivation. This version is a good improvement of the manuscript and I will suggest to publish it.

Only two typographic correction need:

- line 115: JJASin. Missing a space.

- line 150: 2021It. Missing a full stop.